# Does Regulated Deficit Irrigation Affect Pear Fruit Texture by Modifying the Stone Cells?

**DOI:** 10.3390/plants12234024

**Published:** 2023-11-29

**Authors:** Jesús D. Peco, Hava F. Rapoport, Ana Centeno, David Pérez-López

**Affiliations:** 1Departamento de Producción Vegetal, ETSIA—Universidad de Castilla—La Mancha, Ronda de Calatrava, 7, 13003 Ciudad Real, Spain; jesusdaniel.peco@uclm.es; 2Departamento de Producción Agraria, CEIGRAM—Universidad Politécnica de Madrid, Av. Puerta de Hierro 2, 28040 Madrid, Spain; 3Instituto de Agricultura Sostenible (IAS), Consejo Superior de Investigaciones Científicas (CSIC), Avenida Menéndez Pidal s/n, 14004 Córdoba, Spain; hrapoport@ias.csic.es

**Keywords:** *Pyrus communis*, sclereids, water saving, fruit quality, histology

## Abstract

Regulated deficit irrigation (RDI) strategies aim to improve water usage without reducing yield. Generally, irrigation strategy effectiveness is measured as fruit yield, with little consideration of fruit quality. As water deficit and increased plant cell sclerification are often associated, this study explored the effect of RDI on pear fruit stone cells, a crucial trait affecting flesh texture. The presence, distribution, and development of pear fruit stone cells under RDI and full irrigation were compared using *Pyrus communis* L. cv. Barlett trees, employing recently developed microscope image analysis technology. The control treatment was maintained under non-stress conditions, while the RDI treatment received an average of 15% of the control water during the latter part of Stage I fruit development. Observations at the end of Stage I and at harvest revealed no effect on stone cell presence under the RDI strategy tested. The relative area of stone cells within the flesh was greater at Stage I than at harvest, as stone cell expansion occurred early in development, while the (unsclerified) parenchyma cells, a dominant component of the fruit flesh, expanded until harvest. Stone cell cluster density was higher near the fruit core than in the cortex center and exterior. These initial results suggest that well-planned RDI strategies will generally not affect pear fruit stone cell content and, thus, textural quality. Microscope image analysis supported the results from previously used analytical techniques, mainly chemical, while providing a tool for better understanding the process and factors involved in the timing of stone cell differentiation.

## 1. Introduction

Water availability is essential for the profitable production of many woody fruit crops, although with water resources often being scarce or costly, improved water-use efficiency is required [1]. Regulated deficit irrigation (RDI) strategies were proposed as a solution to this challenge, an approach that takes advantage of the varied sensitivity to water stress at different times during fruit growth to minimize the impact of reduced irrigation on yield [2,3]. The RDI technique maintains the tree crop under optimal water status conditions during the major part of fruit development, and only during a phase in which it is considered that crop production is least susceptible to water stress is irrigation reduced until the plants drop to a defined level of water stress.

RDI strategies have been reported as a successful method for pear cultivation without reducing yield by Mitchel et al. [4] and Vélez-Sánchez et al. [5] for European pear (*Pyrus communis*), and Caspari et al. [6], Behboudian et al. [7], and Wu et al. [8,9] for Asian pear (*P. serotine* and *P. bretschneideri*), among others. The deficit period is carried out at a time when fruit growth is least reduced, and if so, can later recover sufficiently so that yield reduction is minimal. Pear fruit growth is driven by the cell division and expansion of the fruit flesh and consists of two periods. In Stage I, the main proportion of cell division takes place, along with cell expansion, which is masked by the concurrent cell division, and in Stage II, only cell expansion occurs [10]. Marsal et al. [11], for example, applied RDI strategies during the latter part of pear fruit development Stage I, finding only a slight tendency for RDI fruits to be smaller. The successful application of the water deficit during Stage I rather than Stage II concurs with the classical analysis by Hsiao [12], suggesting that highly active cell expansion during Stage II generates greater sensitivity to water stress than in Stage I.

In establishing a RDI strategy for the pear tree, however, as in many other fruit crops, successful irrigation management is basically evaluated as yield in size and/or weight and sometimes also includes factors such as dry matter, firmness, and sugars. For example, preliminary results under tropical conditions appear to indicate that RDI strategies did not affect the fruit firmness, pigments, color index, or content of phenols, sugars, or acids [5]. Little attention, however, has been paid to fruit texture, for which stone cell content, a crucial factor contributing to pear fruit quality by negatively effecting texture, has yet to be examined under RDI application.

While the pear fruit flesh is principally composed of parenchyma cells, many stone cells or sclereids are also present. For example, in fruits of *P. communis* at the end of Stage I, up to 35% of the pulp cell number may be stone cells [10]. These cells, a type of relatively isodiametric sclereid known as brachysclereids, are formed via the deposition of a lignified secondary wall within the primary walls of parenchyma cells [13] and are present either in isolated or aggregated groups referred to as stone cell clusters (SCCs) [14,15]. The presence of high concentrations of stone cells elevates the roughness of the flesh texture, reducing economic value [16]. Stone cell formation starts 7–15 days after full bloom (DAFB), and their further development, consisting of additional lignified thickening of the cell wall, continues until approximately 50–65 DAFB, when they reach their maximum size in most varieties [15,17]. Several factors influence the formation of SCCs, including the climate, agricultural practices, and post-harvest handling; however, the tree variety is considered to be the most important [18,19,20,21].

Plant water deficit has been frequently associated with increased cellular lignification [18]. Studies of pear trees have suggested that water stress during the early stages of fruit growth increases the stone cell content in pear flesh, associated with the activity of the peroxidase enzyme (POD), to produce greater cell wall lignification [22]. However, this observed effect is not well known and seems to depend on the tree variety and its resistance, as well as the intensity, duration, and the period in which water stress is suffered [16]. Furthermore, it is important to consider how the different cellular processes (cell division, cell enlargement, and differentiation) may be involved in response to water stress; that is, once a stone cell initiates formation from a parenchyma cell, the onset of the secondary cell wall lignification prevents further cell expansion, although cell differentiation continues as lignification progresses towards the interior of the cell [10,16]. Thus, a water deficit may act by stimulating the initiation of more stone cells or by increasing the lignification of the already initiated cells. Lee et al. [18], basing their observations on fresh- and dry-weight measurements, suggested that a water deficit increased lignin deposition. 

In light of the high potential of water stress to increment sclerification in plant tissues, and the major role of stone cells, produced via sclerification, in determining pear fruit texture, the main goal of this study was to elucidate the effects that a RDI strategy may have on the pear flesh quality, particularly on the stone cell content. We utilized fruits from the experiments of Marsal et al. [11], in which the growing conditions were precisely controlled in young trees growing in large containers. Newly developed microscope image analysis technology was employed to quantify cellular properties, such as the size, development, and distribution of stone cells within the parenchyma cell matrix, providing essential information for understanding and testing the factors involved in stone cell formation. 

## 2. Results

### 2.1. Stone Cells under RDI and Control Irrigation

Characteristics of the SCCs present in the fruit flesh did not differ between the RDI and control treatments (Figure 1); that is, the SCC size, SCC cell number, and SCC cell size showed no significant differences between irrigation treatments at either sampling time. Between sample times, SCC size remained the same (Figure 1A), while stone cell number per SCC decreased, and, in parallel, mean stone cell size in SCCs increased (Figure 1B,C). The percentage of isolated stone cells (found in groups of three or less) was quite small when contrasted with SCCs (Figure 1D), and, although slightly more variable than the other SCC parameters presented in Figure 1, was statistically similar between treatments and sample times. 

The SCC presence in the fruit transverse section, observed as the percentage of tissue area occupied by SCCs and as the number of SCCs per cm^2^, was the same for the control and RDI treatments at each analyzed time (Figure 2). As the area occupied (Figure 2A) and number of SCCs per area (Figure 2B) showed similar patterns, a correlation was performed, indicating a R^2^ value of 0.55. The low correlation may be due to the variability of the data, also indicated by the high errors. Although, as before, no significant differences were found between irrigation treatments, there was a dramatic decrease in both of the SCC presence parameters (area occupied and number) between Stage I and Stage II.

Similarly to the whole flesh analysis (Figure 1 and Figure 2), when the different zones within the fruit flesh were analyzed separately, no significant differences in SCC characteristics (Figure 3) or presence (Figure 4) were observed between the control and RDI treatments within each zone at either time. For several of the parameters studied, however, some differences were observed among zones or between sample times, in spite of the consistent values between treatments.

### 2.2. Stone Cell Dimensions and Presence in Different Fruit Zones

When comparing concentric zones within the fruit, the SCC size presented a pattern of successive decrease from zone In (closest to fruit interior) to zone Ex (closest to fruit exterior). However, neither this pattern nor the specific values obtained in each zone varied between stages (Figure 3A). The number of sclereids per SCC (Figure 3B) also showed a similar pattern of successive decrease among zones. As in the previous case, no significant differences were found between stages. In both cases, while the SCC size and SCC cell number were significantly different between the In zone and Md/Ex zones at Stage I, there were only significant differences between zones In and Ex at Stage II for these parameters. Stone cell size presented lesser differences than SCC size and cell number, but several clear-cut differences were observed among zones (Figure 3C). At Stage I, zone In cells were the smallest, while no significant differences were found between zone Md and Ex cells at both stages. Between stages, an increase in cell size was observed in zone Md but not in zones In and Ex.

Among zones, when the presence of SCCs was measured as percent area, its extent was considerably higher for zone In at Stage I than for the other zones at the same stage; however, no significant differences were found at Stage II among zones. Between fruit development stages, the percent area occupied by SCCs was greatly reduced in zone In, with no significant changes in zones Md and Ex (Figure 4A). In contrast, when SCC presence was viewed as the number per cm^2^ (Figure 4B), values for all three zones were relatively high at Stage I, with higher values observed for zones In and Ex, and lower ones for zone Md. Between fruit development Stages I and II, there was a substantial decrease in the number of stone cell per area in all zones.

### 2.3. Cellular Differentiation in the Fruit Flesh

Differentiation of the SCCs is shown in Figure 5. At Stage I (Figure 5A,B), the stone cells in many of the clusters presented mixed degrees of sclerification, as observed by the different degrees of lignified wall thickening (observed as red with the histological stain utilized in this study); that is, some of the cells forming stone cells (sclereids) presented early stages observed with a thin ring of only slightly lignified walls and most of the cell’s cytoplasm still present. Other cells exhibited successively greater cell wall thickening and a decrease in cytoplasm, ultimately producing an extremely thickened cell wall that fills the entire cell. Notably, the cells positioned farther away from the SCC center showed lower degrees of lignification compared to the central cells. Complete SCC differentiation, observed principally in Stage II, was indicated by full lignification observed in all the stone cells of the cluster (Figure 5C,D).

At Stage II, in addition to being composed of fully differentiated stone cells, several clusters were observed in which parenchyma cells were present within the clusters, seemingly breaking them into multiple groupings (Figure 6A,B). Most of the clusters, however, were exclusively composed of stone cells, as shown in Figure 5C,D.

The sizes of the parenchyma cells with and without contact with SCCs were comparable between irrigation treatments (Table 1). Nonetheless, a minor, though non-significant, tendency was observed for the RDI parenchyma cells to be approximately 10% smaller than those of the control in both Stage I and Stage II (Table 1). Parenchyma cell shape was not influenced by the RDI regime (Table 2).

A considerable, approximately 3-fold increase in parenchyma cell size was observed from Stage I to Stage II for parenchyma cells both without and with contact with SCCs (Table 1). That size increment was even greater in zones In and Md than in zone Ex, and comparable rises were noted in the parenchyma cells around the clusters. With respect to contact with the clusters, it was found that parenchyma cells around the SCCs were substantially smaller than parenchyma cells in all fruit zones and at both stages.

Parenchyma cell shape changed between Stages I and II, becoming less circular (Table 2); that is, Stage I cells were found to be rounder, whereas in Stage II, they were more ellipsoidal. This can also be observed in the cross-section images of pear flesh in Figure 7, in which Stage I parenchyma cells surrounding the cluster are more rounded (Figure 7A) compared to the more ellipsoidal cells in Stage II (Figure 7B). In comparing the parenchyma cells surrounding the clusters with those in the overall fruit flesh, no difference was found at Stage I, but in Stage II, the cluster-surrounding cells were found to be more ellipsoidal than those in the rest of the fruit tissue (Table 2 and Figure 7B). No differences were observed among zones in cell shape.

## 3. Discussion

### 3.1. Effect of RDI Strategies on Pear Fruit Stone Cells

The RDI strategy examined in this study exhibited no discernible impact on the quantity and size of stone cells in pear fruit. Consequently, it can be inferred that one aspect of fruit quality was maintained, which is a fundamental goal of RDI strategies. Despite the presence of numerous studies in the literature that have investigated the effects of RDI strategies on pear cultivation, these studies predominantly concentrated on quantifying water conservation, assessing tree growth, and analyzing fruit size [4,5,6,7,8,11]. There exists a gap, however, in evaluating the potential implications of these strategies on stone cell content, which substantially influences the textural attributes of the fruit. Hence, this study demonstrates the applicability of these strategies in pear cultivation without worsening fruit texture. 

In spite of earlier studies showing an increase in stone cell content as a result of water stress [18], we believe that, in our specific case, the application of stress after the critical phase of the initial formation of sclereid primordial cells, along with the relatively mild intensity of the administered water, may have collectively contributed to the preservation of pear texture quality. 

Sterling et al. [23] established that individual sclereids are initiated within the cortex of *Pyrus communis* cv. Barlett pears one–two weeks after bloom. Later, until 8 weeks after full bloom, the differentiation of new stone cells mainly occurs in the parenchyma surrounding the previously formed sclereids, forming clusters. Consequently, the quantity of SCCs is determined via the number of individual sclereids, formed during the very early stages of fruit growth [16]. The period of water stress we applied encompassed the interval from 32 DAFB to 60 DAFB, after the initial formation phase of sclereid cell initiation; that is, the early determination of individual stone cells would produce the similar SCC number we found between the control and RDI treatments, even though continued lignification and further stone cell differentiation would continue. As a result, we can affirm that in establishing RDI strategies, understanding the critical stages of fruit cell differentiation is crucial to prevent detrimental impacts on fruit quality.

Regarding the size of SCCs, experiments involving drought stress in *Pyrus pyrifolia* cv. niitaka demonstrated that intense water stress periods from 30 DAFB to 60 DAFB led to an increase in stone cells, quantified in terms of dry weight, in comparison to control fruits [18]. Those authors suggested that the increase in stone cells could be attributed to the decrease in leaf water potential, which hindered the absorption of calcium. As a result of this imbalance, there was an increase in peroxidase enzyme activity (POD) [18]. High POD activity is known for its role in promoting stone cell formation within the flesh, primarily due to the heightened cell wall lignification [22]. POD enzyme activity increases in response to stress factors, such as drought, producing augmented cell wall lignification [22,24]. This phenomenon can potentially be directly associated with the heightened stone cell presence in the pear fruit. However, it is essential to consider that the intensity of water stress in that experiment was considerably more severe than in our own, reaching leaf water potential values of -2.7 MPa [18], whereas the lowest potential peak in our study was −1.7 MPa [11]. This implies that our plants may not have experienced a sufficiently pronounced stress level to trigger an elevation in POD enzyme activity and subsequently enhance the SCCs density per unit area. As a result, we agree that meticulous control of the water stress intensity applied to trees within RDI strategies is imperative to avoid high levels that could adversely impact fruit texture quality, as advocated by Torrecillas et al. [25].

### 3.2. Stone Cell and Cluster Development

Within the SCCs, differences may be observed among the stone cells in their degree of differentiation. The pear fruit receptacle at bloom is composed of parenchyma cells, the majority of which will divide and expand during development to produce growth in fruit size, while some of them will differentiate into sclerified stone cells. In stone cell differentiation, certain parenchyma cells develop a highly specialized secondary cell wall that thickens due to lignin deposition towards the interior of the cell, a process which continues progressively until the entire cell wall is fully thickened [10,16,26]. In Stage I SCCs (Figure 5A,B), one can observe mature, fully lignified stone cells, which are completely stained red for lignin, and others, less developed, in which the lignification is still in progress and appears as a ring. Cheng et al. [16] designated those as partially lignified cells, in which a lignified ring surrounds the shrinking cell lumen as sclereid primordia. Within the Stage I SCCs, the fully mature sclereids are observed in the centers of the clusters, while the partially developed sclereid primordia are found towards the cluster exterior. These observations agree with the interpretation by Sterling [23] that clusters are built as newer sclereids form around the earlier differentiated ones.

In the mature SCCs in Stage II (Figure 5C,D), the central stone cells in the cluster tend to be smaller than those more towards the cluster’s exterior. Once a stone cell begins to sclerify, its division and expansive growth stop due to the rigidity of the secondary cell wall [26] and the degradation of vesicles and organelles present within these cells [14,19]. Since the growth of stone cells ceases following the onset of lignification, the larger stone cells would seem to develop from larger parenchyma cells, present at a later moment in fruit expansion, providing further evidence of the progressive formation of stone cells outwards from the center of the cluster [23]. Moreover, the average stone cell size increased from Stages I to II (Figure 1C and Figure 3C, respectively), also indicating the differentiation of sclereids from larger parenchyma cells formed later in fruit growth, as well as the occurrence of some lignification later than Stage I. This phenomenon was only evident, however, in the middle zone of the flesh fruit. It could be possible that stone cells situated more externally and internally in the fruit flesh completed their formation earlier than those located in the middle of the fruit flesh. This, however, would contradict Esau et al. [26], who described SCC formation as beginning in the deeper regions of the pear flesh and progressing outward.

A slight reduction in the number of stone cells per SCC in Stage II compared to Stage I was found (Figure 1B). This observation suggests that, in this pear variety, certain stone cells might become detached from the SCC, potentially occurring via invasive parenchyma growth, as can be observed in the microscopic images (Figure 6). This finding contradicts the previously postulated pattern of development, in which stone cell aggregation intensifies throughout the fruit’s development [16]. Nonetheless, it is important to note that such instances of detachment were relatively uncommon, since the number of isolated stone cells encountered in the fruit pulp was significantly smaller compared to the number of stone cells grouped within SCCs. The reduction in SCC size might hold significance due to the intrinsic connection between the texture of the fruit flesh and the size of the SCCs [16]. In this manner, SCCs with diameters exceeding 250 µm yield a coarse and gritty flesh texture; those ranging between 150 µm and 250 µm offer a soft texture with a slight gritty sensation, and those less than 150 µm result in a highly soft texture [16,17]. Consequently, the separation of isolated stone cells from the SCC could possibly contribute to an improvement in the texture quality of the pear but requires further research. 

Limited information is available regarding the comprehensive characterization of SCCs within different regions of the fruit pulp, likely related to the required time and available analytical techniques. Prior studies have convincingly demonstrated a higher overall quantity of SCCs in the inner fruit region across five pear cultivars [17,27], with only one variety showing a prevalence near the peel [17]. Our findings in Stage I concur with these previous studies, indicating a substantial occurrence of SCCs in the inner zone, in contrast to the middle and exterior regions of the fruit. This observation could be attributed to their larger size and a higher number of stone cells per SCC in the inner fruit. This pattern coincides with the findings of Tao et al. [27] and Li et al. [17], who reported that SCCs in the inner region were notably larger than those found in the remaining flesh. However, at fruit maturity (Stage II), our results indicate that the area occupied by SCCs and their number per cm^2^ are similar across all zones.

### 3.3. Relationship between Parenchyma Development and SCCs

From Stage I to Stage II, a substantial drop to almost 20% was found in the number and density (percent area) of SCCs (Figure 2). The decrease in the relative presence of SCCs results from the interaction between stone cell differentiation and parenchyma cell expansive growth during fruit development; that is, on the one hand, it has been shown in many studies and in a large number of different varieties that the pear fruit SCC content stabilizes around 50–65 DAFB [15,17,21], after which the stone cells remain static without growth and are not degraded after formation [14,16,28]. Secondly, cell expansion in parenchyma cells [11,17,19] continues throughout pear fruit growth, diluting the SCC density in the flesh fruit. In addition to the generally high SCC presence observed for the complete fruit at Stage I (Figure 2), we also noted a more pronounced presence in the inner zone (Figure 4). Then, at Stage II, the decrease in percent occupied area was greater for the inner zone than for the rest of the flesh. Interpreting these differences, however, requires further information regarding the timing of developmental patterns in the different fruit zones. Our results, using the image analysis of histological preparations, concur with the acknowledged dilution pattern of SCCs within the pear flesh parenchyma matrix, and additionally provide further quantification and details of cellular differentiation and spatial differences within the fruit.

The microscopic images (Figure 7) and the shape analysis (Table 2) of the fruit flesh parenchyma cells revealed a pattern in which these cells lose roundness and become more irregular during expansive growth. This phenomenon has been previously described in pear fruits [14,19], as well in others such as olives [29], and could be precisely described using image analysis. In contrast to the irregularity in parenchyma cell shape, the stone cells maintain more rounded shapes due to the cessation of growth when their lignified secondary wall is first established in young and still rounded parenchyma cells and the structural rigidity provided by further wall growth and lignification [19]. Furthermore, as stone cells initiated their formation early in fruit development, when the parenchyma cells from which they were differentiated were smaller and rounder, this timing undoubtedly influenced their geometry. 

During fruit growth, the shape of parenchyma cells whose cell walls were contiguous with the SCCs became notably more elongated than the rest of the parenchyma cells, radiating outwards from around the SCCs. This event was previously described by Nii et al. [14], and appears to be a consequence of internal physical pressures during fruit expansion; that is, the lack of expansion by the lignified SCC cells restricts the possible directions for parenchyma cell growth within the fruit cell matrix. We also found that the parenchyma cells touching the clusters were larger than the other parenchyma cells at fruit maturity (Table 1). Larger parenchyma cells would likely indicate a lower presence of cell walls, suggesting softer fruit flesh immediately surrounding the SCCs, potentially producing a greater sense of disagreeable gritty texture to the fruit consumer.

### 3.4. Microscope Image Analysis Techniques for Advancing SCC Studies

Studying the SCCs in pear flesh presents some difficulties regarding how to best choose useful parameters to measure and evaluate characteristics such as their presence, formation, and distribution, among others. In general, the evaluation of SCC content in pears has relied on separation techniques and the measurement of sclereid content weight, or by assessing the lignification process within the fruit [18,19,21,27,28,30,31,32]. While these methods have effectively revealed variation in SCC content among different varieties or in response to specific treatments or environmental factors, they offer limited insights into the developmental process of SCCs within the fruit. Conversely, studies utilizing microscope techniques have a relatively high cost–benefit ratio mainly due to the considerable time they require. These studies can be categorized into two groups: those that provide a descriptive view of SCC development [10,14,18,19,26,33], and those that employ histology for characterization, solely focusing on SCC content and size [15,17,20,22]. The quantitative measurements derived from microscopic analyses necessitate the acquisition of a large number of images, which are subsequently laboriously analyzed manually to obtain precise data. One promising solution to this challenge is the utilization of image processing programs, which offer rapid, cost-effective, and highly precise image analysis. In our study, the implementation of these automatic programs has enabled the examination of a substantial number of images, ensuring the acquisition of precise information of a quantitative nature. Indeed, the use of image analysis helps form a bridge between anatomical observation and quantitative analytical techniques. This not only facilitates accurate confirmation or refutation of previously established knowledge but also provides deeper insights into the development of SCCs within the pear flesh.

## 4. Materials and Methods

### 4.1. Growth Conditions and Irrigation Treatments

Two-year-old pear trees of the *Pyrus communis* Williams variety were planted in 120 L containers. The use of pots with equal soil volume available for root growth allowed for careful control and monitoring of tree water status and similarity within treatments, avoiding the inherent heterogeneity of soils and occupied soil volume. Once established in the pots, the trees were carefully selected for similar vegetative growth and, later, fruit load.

During the first year of growth in the containers (year three), the limit of plant water stress for local conditions was established as −0.8 MPa using stem water potential (Ψ_stem_) measured with a plant water status console (Model 3005, Soil Moisture equipment Corporation, Santa Barbara, CA, USA). Additionally, photosynthetic parameters were measured with the IRGA system to ensure that the plants were maintained under optimal water status conditions. In the following two years after planting, two irrigation regimes, control and regulated deficit irrigation (RDI), were tested. For the control treatment, irrigation was carried out to provide non-stress conditions in the plants, maintaining the average Ψ_stem_ around −0.8 MPa (as in the previous growing years). RDI plants were exposed to a deficit irrigation period within the latter part of fruit growth Stage I (32–60 DAFB), in which the applied water was gradually reduced until the values of Ψ_stem_ reached around –1.4 MPa. More details regarding crop growth conditions have been described by Marsal et al. [11], as are measurements of parameters such as fruit volume, fresh weight, percent dry matter, and soluble solids.

### 4.2. Fruit Tissue Sampling and Preparation

In the second year of the RDI experiment, when the trees were five years old, fruits were collected at the end of Stage I and Stage II (at harvest). The Stage I harvest was at 60 DAFB (days after full bloom), corresponding to the end of Stage I and the beginning of rehydration via irrigation. The stage II harvest was at 120 DAFB, corresponding to the end of Stage II and commercial harvest time. At Stage I, 9 fruits/treatment (3 fruits per tree in 3 trees per treatment) were taken, while at Stage II, 4 fruits/treatment (2 fruits per tree in 2 trees per treatment) were sampled. A 5 mm transverse slice at the widest portion of each fruit was taken, and the slices were fixed in FAE (formalin: acetic acid: 60% ethanol, 2:1:17, *v*/*v*/*v*) and preserved in 70% ethanol. Standardized portions of the fruit cortex were identified by first staining the complete slices for 2–3 h in 70% ethanol with weak toluidine blue [34] to show the different vascular bundles (associated with sepals, petals, and carpels) present in pome fruit. One (in Stage I) and two (in Stage II) wedge-shaped radial sectors extending from the sepal vascular bundle (SB) to the fruit exterior were cut per fruit slice (Figure 8). The sectors were dehydrated in tertiary butyl alcohol and processed according to standard paraffin procedures, obtaining microtome sections of 10–12 µm, which were stained with tannic acid, iron chloride, safranin, and fast green, adapted from Johansen [35] and Ruzin [36].

### 4.3. Histological Assessment

For histological evaluation, the fruit sectors were divided at equal distances along the radius into three concentrically oriented zones: the area closest to the fruit exterior (zone Ex), the intermediate, or middle zone (zone Md), and the area closest to the fruit interior (zone In), as depicted in Figure 8. Two square images per zone, each 0.06 cm^2^, were observed in each sector for Stage I, and six images per zone were obtained for Stage II. Microscopic images were acquired with an Olympus BX51 light microscope and recorded using a digital camera (Olympus SC50) attached to the microscope using CellSens Imaging Software (V.5.10, Olympus Corporation, Tokyo, Japan). Finally, 54 images per treatment were analyzed in Stage I and 144 in Stage II. All parameters were first determined for the three different fruit flesh zones, and then overall (globally) for the total area studied. The global values were calculated by considering the proportional area contribution of each zone.

Individual stone cells and stone cell clusters were observed. Stone cell clusters, referred to as SCCs (following the nomenclature of Nii et al. [14]), were considered as such when the cluster contained more than three adjoining stone cells. Individual stone cells and those found in groups of two or three, in all cases separated from other stone cells by at least one parenchyma cell, were designated as isolated. Given the relatively small number of isolated stone cells (Figure 1D), the majority of the stone cell analyses only utilized the SCCs. A macro was developed for use with Image J software (V.2.1.0, National Institutes of Health, Bethesda, USA) to automatically determine the size of the SCCs, the number of SCCs and isolated stone cells, the area occupied by the SCCs, and the number of SCCs per area. In addition, the average stone cell size was calculated from the data. Data automatically determined via software were compared with data obtained via manual determination, achieving a high correlation (R^2^ = 0.9). 

The parenchyma cells were divided into two types: those with and without direct contact with stone cells, and a second Image J software (V.2.1.0, National Institutes of Health, Bethesda, USA) macro was established for measuring parenchyma cell size and circularity index. The internal size of each cell was identified and measured, and the average cell size was subsequently calculated. A circularity shape index ranging from 0 to 1 was used to assess the roundness of the cells, in which values closer to 1 indicate a more circular shape and values closer to 0 imply a more elliptical shape. To obtain this value, the largest and smallest diameters of each parenchyma cell were automatically measured, and the following equation was used:Mesocarp cell circularity = Largest diameter/Smallest diameter

### 4.4. Statistical Analysis

All the data reported were tested for homogeneity of variance, normal distribution, and sphericity, and then were analyzed via a nested analysis of variance with repeated measures (zones nested with fruits) followed by the Tukey test to determine the significant differences (*p* ≤ 0.05). In two analyses that considered percentages, that is, (1) the percentage area occupied by SCCs, and (2) the ratio of the isolated stone cell number to the number of stone cells in SCCs, the data were subjected to arcsine transformation before ANOVA. Statistical analysis was performed using an R package.

## 5. Conclusions

The RDI (regulated deficit irrigation) strategy examined in this study had no discernible impact on the quantity and size of stone cells in pear flesh. Consequently, it can be reasonably inferred that one of the fundamental goals of RDI strategies, which is to maintain fruit quality, has been successfully achieved. However, we emphasize the importance of considering the timing, intensity, and duration of water stress application during RDI strategies to avoid potential adverse effects on the textural quality of pears.

We employed automated image analysis programs, enabling us to conduct a more comprehensive analysis of the SCC (stone cell cluster) characteristics in the fruit flesh. This study included examinations of the quantity and size of stone cells within the SCCs, previously unfeasible using conventional tools. With these new data, we have not only been able to validate or refute existing knowledge with more accurate information derived from a substantial number of images and measurements but have also acquired deeper insights regarding the development of pear fruit sclereids.

The authors believe that the information presented in this research can be used to develop more effective RDI strategies in pear cultivation to protect fruit quality, as well as to improve the understanding of SCC development in pear fruit flesh.

## Figures and Tables

**Figure 1 plants-12-04024-f001:**
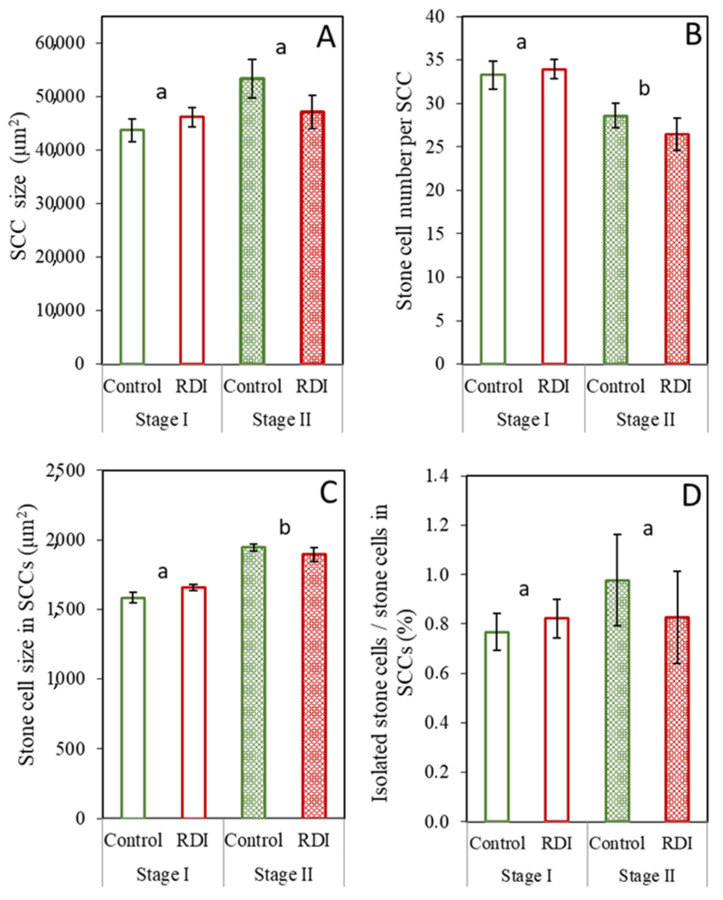
SCC attributes across the complete fruit flesh area: SCC size (**A**), stone cell number per SCC (**B**), stone cell size in SCCs (**C**), and percentage of isolated stone cells compared to the stone cells within SCCs (**D**). Treatments were in the form of two irrigation regimes [control and regulated deficit irrigation (RDI)] sampled at two different times (Stages I and II). Error bars represent the standard error. No significant differences were found at *p* ≤ 0.05 between treatments (Tukey test). Values with different lowercase letters are significantly different between stages at *p* ≤ 0.05 (Tukey test).

**Figure 2 plants-12-04024-f002:**
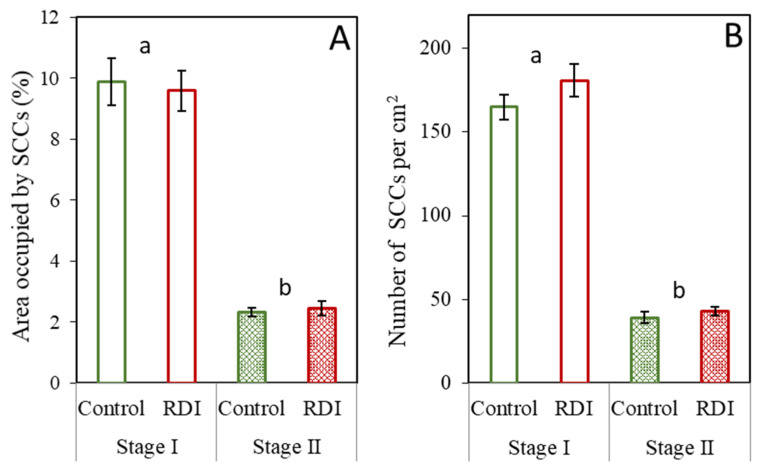
Presence of SCCs in fruit transverse sections determined as the percentage of area occupied (**A**) and the number of SCCs per cm^2^ (**B**). Observations of pear fruit under two irrigation regimes [control and regulated deficit irrigation (RDI)] at two different times (Stages I and II). The error bars represent the standard error. No significant differences were found at *p* ≤ 0.05 between treatments (Tukey test). Values with different lowercase letters are significantly different between stages at *p* ≤ 0.05 (Tukey test).

**Figure 3 plants-12-04024-f003:**
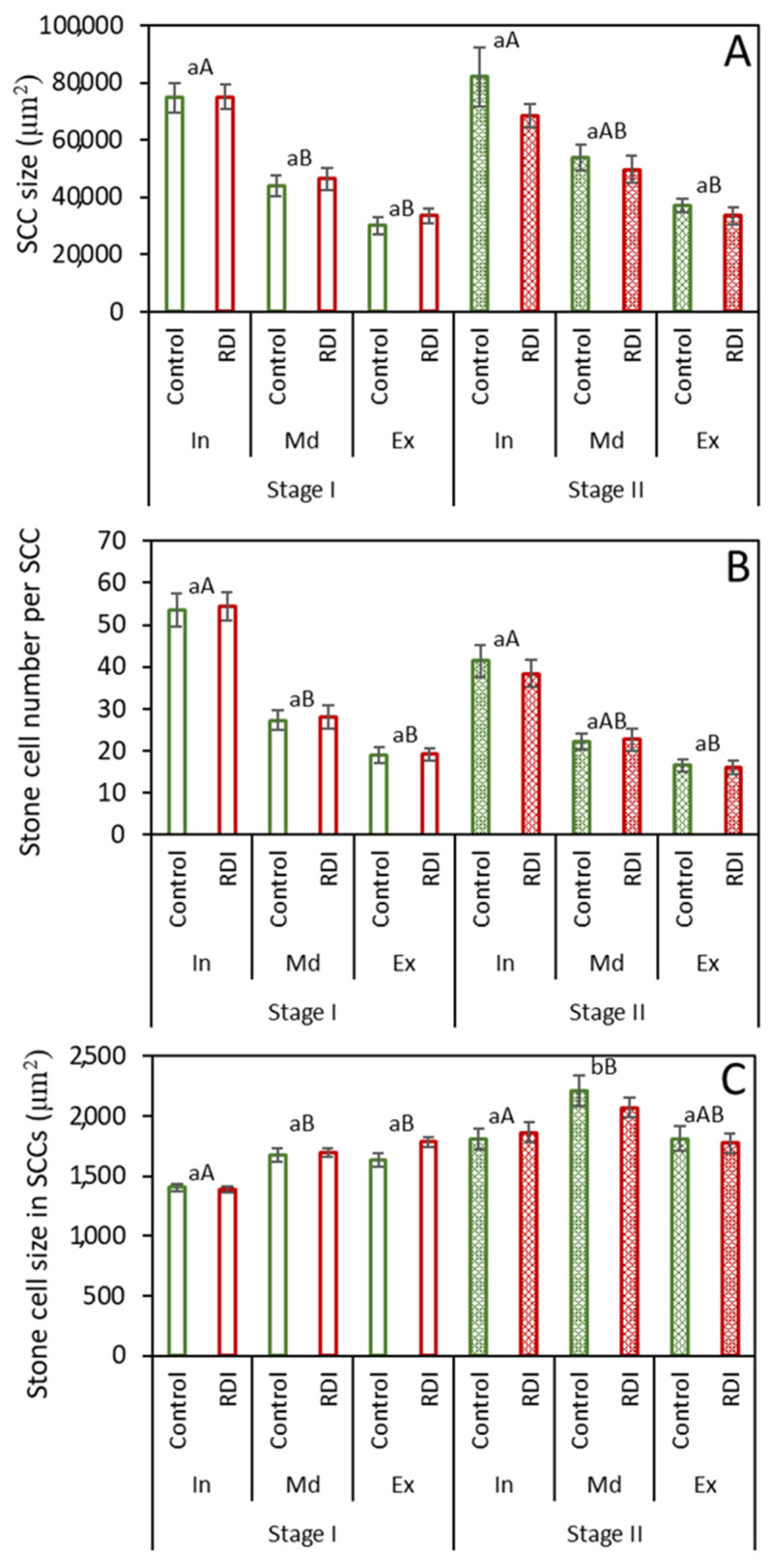
SCCs and SCC cell attributes in separate concentric zones in transverse slices of pear: SCC size (**A**), stone cell number per SCC (**B**), and stone cell size in SCCs (**C**) under two irrigation regimes [control and regulated deficit irrigation (RDI)] at two times (Stages I and II). Zone In is closest to the fruit center, zone Md denotes an intermediate position, and zone Ex is closest to the exterior. Error bars represent standard error. No significant differences were found at *p* ≤ 0.05 between treatments (Tukey test). Values with different lowercase letters were significantly different between stages, and values with different capital letters were significantly different between zones at *p* ≤ 0.05 (Tukey test).

**Figure 4 plants-12-04024-f004:**
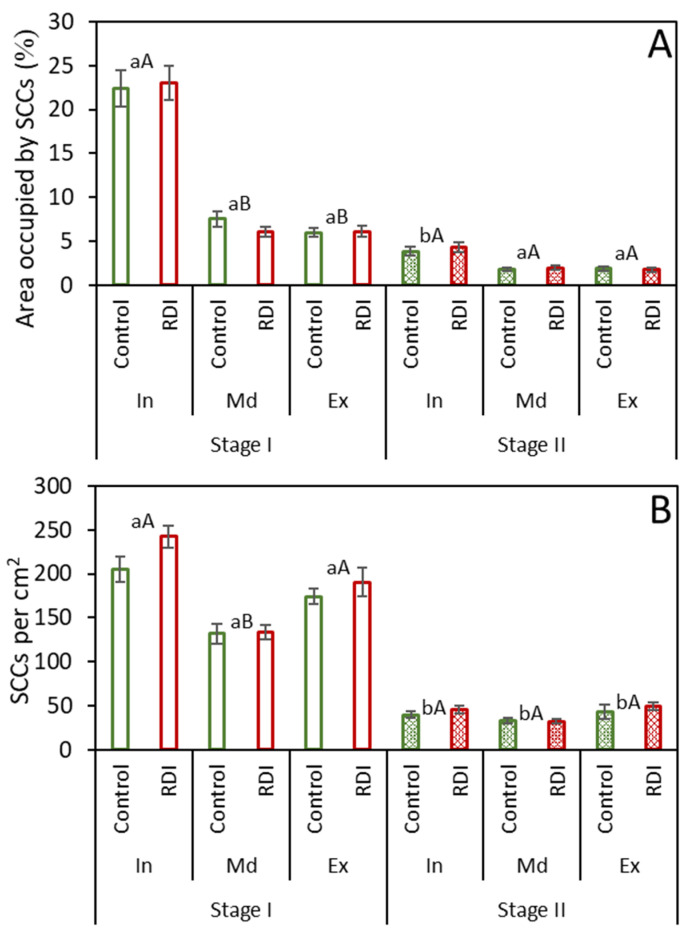
Presence of SCCs in concentric zones of pear fruit transverse sections. Data are presented as the percentage of area occupied by SCCs (**A**) and the number of SCCs per area (**B**) under two irrigation regimes [control and regulated deficit irrigation (RDI)] at two different times (Stages I and II). Zone In (interior) is nearest to the fruit center, followed by zone Md (middle), and Zone Ex (exterior) is closest to the fruit exterior. Error bars represent the standard error. No significant differences were found at *p* ≤ 0.05 between treatments (Tukey test). Values with different lowercase letters were significantly different between stages, and values with different capital letters were significantly different between zones at *p* ≤ 0.05 (Tukey test).

**Figure 5 plants-12-04024-f005:**
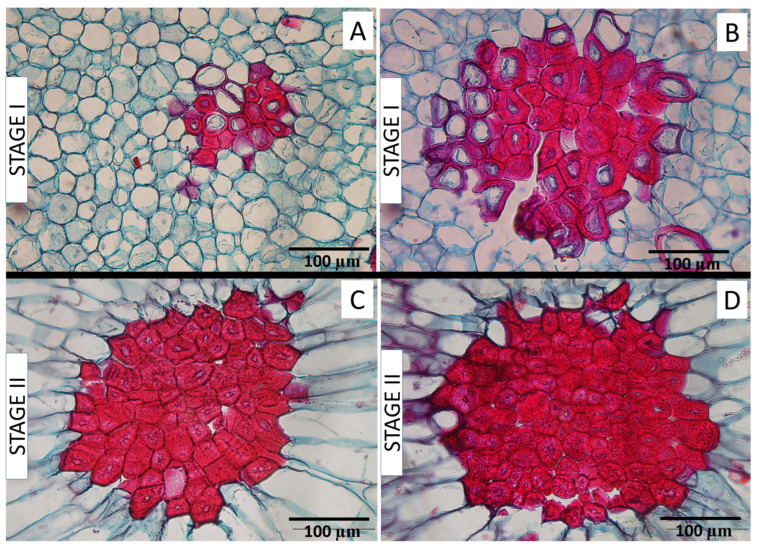
SCCs in transverse sections of the cortical flesh of pear fruit presenting varied degrees of differentiation of the clusters and their component cells. Mixed degrees of cell wall sclerification (lignification; red stain) observed in Stage I (**A**,**B**), and complete sclerification of all cluster cells in Stage II (**C**,**D**).

**Figure 6 plants-12-04024-f006:**
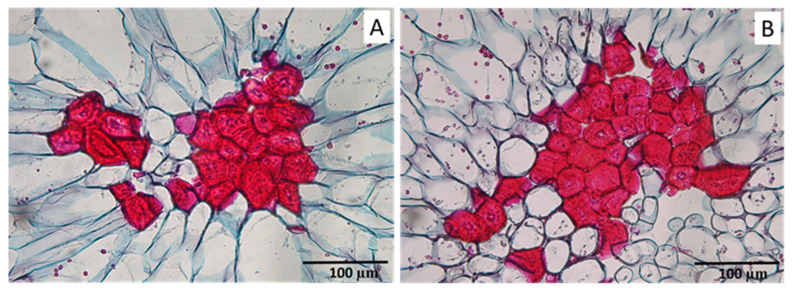
Stone cell clusters in which there are stone cells separated from the main SCC by parenchyma cells. (**A**,**B**) Images of two different clusters from the control treatment, observed in Stage II. Parenchyma cells are visualized with blue-stained cell walls, while SCCs are stained in red.

**Figure 7 plants-12-04024-f007:**
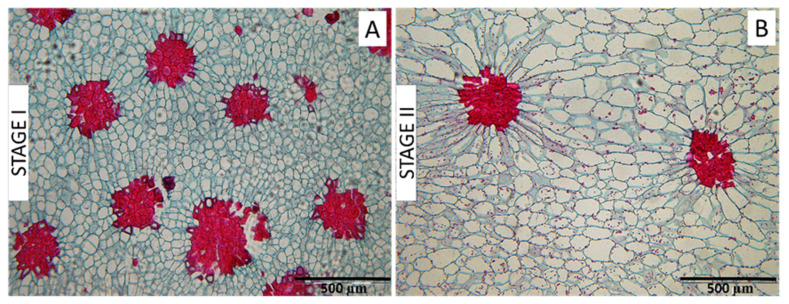
Variability in parenchyma cell shape based on fruit development and spatial relationship with SCCs. Images were obtained in the control treatment of Stage I (**A**) and Stage II (**B**). Parenchyma cells are visualized with blue-stained cell walls, while SCCs are stained in red.

**Figure 8 plants-12-04024-f008:**
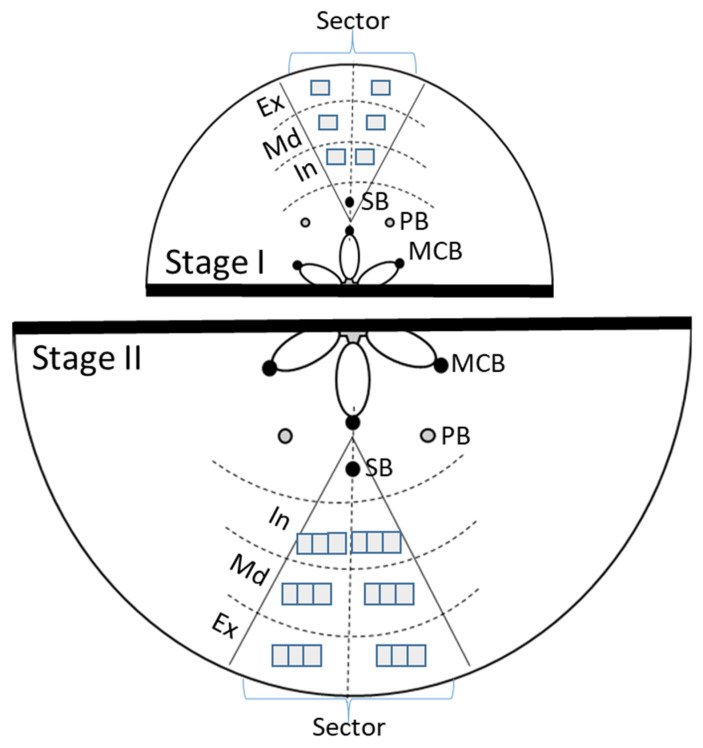
Scheme for the analysis of pear fruit cells in sections of Stages I (upper) and II (lower). Wedge-shaped sectors were cut from central transverse fruit slices, each extending from a sepal bundle (SB) to the fruit exterior, and processed for histological study. For measurements, the sector was visually divided into two halves and three concentrically oriented zones: Ex (Exterior), Md (Middle), and In (Interior). The squares within each zone represent the number of microscope image areas (0.06 cm^2^) captured and analyzed (two images per zone for Stage I; six per zone for Stage II). SB: sepal bundle, PB: petal bundle, and MCB: median carpelary bundle.

**Table 1 plants-12-04024-t001:** Parenchyma cell sizes in pear fruit flesh at Stages I and II for two irrigation treatments [control and regulated deficit irrigation (RDI)]. The transverse sections were measured in three concentric zones (In, interior; Md, middle; and Ex, exterior), and the parenchyma cells were divided into general (all parenchyma cells except those in contact with stone cells) and around cluster (those in direct contact with the stone cells). Values show the means. No significant differences were found at *p* ≤ 0.05 between treatments (Tukey test). Values with different lowercase letters were significantly different between stages, and values with different capital letters were significantly different at *p* ≤ 0.05 between types of parenchyma cells (Tukey test).

			Size (µm^2^)			
		Zone	Parenchyma Cells (General)		Parenchyma Cells around Cluster	
Stage I	Control	In	2147 ± 67.51	aA	1853 ± 89.44	aB
		Md	3034 ± 109.95	aA	2763 ± 116.29	aB
		Ex	3154 ± 42.84	aA	2772 ± 62.40	aB
	RDI	In	1879 ± 49.62	aA	1787 ± 66.86	aA
		Md	2956 ± 99.95	aA	2691 ± 126.30	aB
		Ex	2889 ± 108.04	aA	2558 ± 125.96	aB
Stage II	Control	In	9472 ± 438.13	bA	7985 ± 411.44	bB
		Md	13,435 ± 528.35	bA	12,560 ± 349.52	bB
		Ex	13,139 ± 489.40	bA	11,724 ± 494.09	bB
	RDI	In	8706 ± 178.79	bA	7276 ± 225.49	bA
		Md	13,400 ± 428.39	bA	11,143 ± 483.39	bB
		Ex	11,327 ± 570.13	bA	9658 ± 373.16	bB

**Table 2 plants-12-04024-t002:** Parenchyma cell shape in pear fruit flesh, using a circularity index on a scale ranging from 0 to 1. A value closer to 1 indicates a more circular shape (further details in the Materials and Methods section). Data from Stages I and II for two irrigation treatments [control and regulated deficit irrigation (RDI)]. The parenchyma cells were divided into general (all parenchyma cells except those in contact with stone cells) and those in direct contact with the stone cells. Values show the means. No significant differences were found at *p* ≤ 0.05 between treatments (Tukey test). Values with different lowercase letters were significantly different between stages, and values with different capital letters were significantly different at *p* ≤ 0.05 between types of parenchyma cells (Tukey test).

		Circularity (1–0)	
		Parenchyma Cells (General)	Parenchyma Cells around Cluster
Stage I	Control	0.67 ± 0.01 aA	0.67 ± 0.01 aA
	RDI	0.62 ± 0.01 aA	0.68 ± 0.01 aA
Stage II	Control	0.58 ± 0.01 bA	0.42 ± 0.01 bB
	RDI	0.55 ± 0.01 bA	0.45 ± 0.01 bB

## Data Availability

Data are available at: https://edatos.consorciomadrono.es/dataset.xhtml?persistentId=doi:10.21950/JXDLTN (accessed on 15 September 2023).

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
