# Peer review of "Does Regulated Deficit Irrigation Affect Pear Fruit Texture by Modifying the Stone Cells?"

_plants, 2023, doi:10.3390/plants12234024_

Round 1
Reviewer 1 Report
Comments and Suggestions for Authors
The overall design of the article is not reasonable, the experiment only involves the morphological observation of stone cells, the title "pear fruit texture" is not reflected, the research content is not deep enough. The writing is not careful and the diagrams are not standardized, so it is recommended to resubmit the article.
Major issue:
1、 The article is not clear and focused. The scientific question of the paper is "Does the RDI strategy affect pear stone cells", but the overall description of the article focuses on comparing the differences in pear fruit between Stage I and Stage II, and seldom mentions the effects of the two irrigation methods on pear fruit.
2. The Introduction describes the RDI strategy as an attempt to improve water use efficiency, but in the experimental design the RDI plants were irrigated normally during the usual time and the irrigation was reduced only during the later stages of fruit development and growth, which is different from deficit-regulated irrigation in actual production.
3. The article has too little control analysis between RDI and Control, mainly focusing on the comparison between Stage I and Stage II. Are the pears used in Figure 7 and Figure 8 from the control or experimental group?
Tiny issue:
1. The language in the article is not rigorous, please check it carefully. 52-56 lines are repetitive; Stage II corresponds to what period of the fruit? What is the specific time of harvesting for Stage I and Stage II in the results?
2、 Figure 6 What is the difference between Figures A and B, and Figures C and D. Figure 7 Difference between Figures A and B?
3、 Where are the data errors in Table 1 and 2? Why don't we use bar charts?
4、 Please describe the specific measurements in row 99 and where are the corresponding results?
5、 Is the plant material used in row 103 a four or five year old pear tree?
Comments on the Quality of English LanguageThere are sentences in the article that are hard to understand, so please revise them carefully.
Reviewer 2 Report
Comments and Suggestions for Authors
The paper describes the traits associated to stone cells in two stages of pear development, one watering treatment and areas withing the pear fruit. The traits include abundance and a comprehensive morphological characterization. I complement the authors on the paper, it is well written and easily followed, I thoroughly enjoyed the chance to review it. There are a couple of main issues that I think should be addressed and some minor ones that I will describe further on.
One of the issues that must be addressed is statistical. As samples were taken from within one fruit, these observations are not independent form on another. As such the analysis should have included a nested component of samples (inner, middle outer) within fruits. I’m not sure there are enough degrees of freedom from the number of samples collected. The second issue that must be addressed is the use of digital data. One of the arguments used in favor of automated methods is their ease of use to process large amounts of data and in theory reduce bias. This is true when the program or script used was calibrated such that it reflected an equivalent value (with some error of course) of what an observer would have seen. Was any form of calibration made on the image analysis part? If so, please add a small explanatory paragraph. The third issue is that the authors concentrated on one component of Pear quality and not others. Even though RDI can be beneficial in mot having an important effect on stone cells it may influence water or sugar content, fruit size and or weight of other traits associated to fruit quality. A small paragraph discussing other consequences of RDI might highlight the need for a more comprehensive study that includes SSCs as a factor in different RDI methodologies.
Minor comments.
L14 plant cell sclerification are often
L15. A crucial trait
L19 15% of the control
L21 RDI strategy tested
L27 mainly chemical
L27 better understanding the process and factors involved in the timing of stone cell differentiation.
L33 which required
L35 challenge, an approach that takes…
L42 can later recover
L43 yield reduction in minimal
L55 along with cell expansion…
L57 during the latter part
L59 successful application of water deficit
L61 delete “in that period”
L62. In establishing an RDI….
L61-66. Aren’t size, and weight also traits of fruit quality
L64-66 repeated sentence
L101. With a pressure chamber (please add models of all apparatus used)
L207. Presented smaller differences
L265 showed smaller degrees of
L333 it can be inferred that one aspect of fruit quality was maintained which is a fundamental goal of RDI strategies. Despite the presence of…
L341 Bit there may be other effects that were not quantified in this paper such as fruit weight, size, water and sugar content, among others.
L349 stone cells mainly occurs in the…
L369 consider that water stress
L380 delete “as shown in figure 6”.
L456 revealed a pattern..
Figure 3. I would expect both area occupied and number of SSC to be highly correlated please test.
Comments on the Quality of English LanguageMinor issues included in the above review
Reviewer 3 Report
Comments and Suggestions for Authors
The topic of the manuscript is very interesting and in many parts it is well written. I have just some advice and some curiosity:
1) How many trees were studied in this trial?
2) Did you cover the 120L containers with anything to avoid soil evaporation?
3) Did you detect soil humidity?
4) In M&M you write that you have harvested and analyzed 2-4 fruits in 4 control trees and in 4 stressed trees during I stage and then 2-4 fruits in 2 control trees and in 2 stressed trees during II stage. Did you evaluate fruits of same trees? I mean, did you study the same trees in I and II stage? Or not?
5) Should be possible to see a small and simple map of the experimental design? Just to see the position of control and stressed containers (and plants).
Thank you.
Reviewer 4 Report
Comments and Suggestions for Authors
This study investigates the impact of Regulated Deficit Irrigation (RDI) on pear fruit stone cells, which play a crucial role in fruit texture. RDI aims to optimize water use without compromising yield, with a primary focus on fruit yield rather than quality. The study found that the RDI strategy did not significantly affect the presence of stone cells, and stone cell expansion primarily occurred early in fruit development. Additionally, the research revealed that stone cell clusters were more densely concentrated near the fruit core. In summary, this study suggests that well-planned RDI strategies are unlikely to influence pear fruit stone cell content and, consequently, textural quality.
The study topic is very interesting. The relationship between irrigation and fruit quality is an important area for horticultural science. A weakness is the pot experiment, which limits the significance of the result. I very much appreciate the methodology of the experiment and the good and thorough discussion. I consider the paper to be very successful.
Specific comments:
- Use scientific terms in the title and replace "stone cells" with "sclereids" or "sclerenchymatous meshes".
M+M
- Is it possible to quantify the water savings of RDI?
Reviewer 5 Report
Comments and Suggestions for Authors
"The manuscript titled "Does regulated deficit irrigation affect pear fruit texture by modifying the stone cells?" represents an original research experiment that significantly contributes to the field of plant sciences. The primary objective of the study was to investigate the impact of regulated deficit irrigation (RDI) strategies on pear flesh quality, specifically focusing on stone cell content. Although the authors conducted the experiment in controlled conditions (mentioned briefly as being in containers), specific details about these conditions were not provided. The histological assessment was performed using a microscope.
The manuscript, fitting well within the scope of the Plants journal, presents compelling studies and results that enhance our understanding of the topic. The introduction is well-composed, and the materials and methods section includes essential details about experimental preparations and analyses. However, there is a notable omission regarding the growth conditions, which should be addressed.
The data analysis is generally adequate, and the results offer valuable insights. The authors discuss the obtained data sufficiently.
Despite these strengths, there are several shortcomings in the manuscript that require correction before publication:
1) Authors did not make a research hypothesis.
2) Keywords: they cannot be this same as in title.
3) MM section: where did the plant material come from?
4) As I mentioned above, in my opinion the authors must provide the conditions of the experiment. It is true that they quote work in which such an experiment has already been carried out, but this does not exempt them from providing basic information (what was the temperature and humidity of the air, what was the light - intensity and proportion of day and night). Only details about the container itself may be quoted.
5) MM section: authors should specify the exact location of the experiment
6) MM section: in this version the subsection about statistical procedure is titled: Fruit tissue preparation….?
7) Information about n – number of repetitions should added in tables/figures captions and in MM section.
8) MM section, statistical analysis: have all the assumptions (normality of distribution and equality of variances) been met to be able to use the ANOVA test?
